# A gateway conspiracy? Belief in COVID-19 conspiracy theories prospectively predicts greater conspiracist ideation

**Javier A. Granados Samayoa**[1], **Courtney A. Moore**[1], **Benjamin C. Ruisch**[2], **Shelby T. Boggs**[1], **Jesse T. Ladanyi**[1], **Russell H. Fazio**[1] *

1 Department of Psychology, The Ohio State University, Columbus, Ohio, United States of America,
2 School of Psychology, University of Kent, Canterbury, United Kingdom

* fazio.11@osu.edu

## Abstract

A primary focus of research on conspiracy theories has been understanding the psychological characteristics that predict people's level of conspiracist ideation. However, the dynamics of conspiracist ideation—i.e., how such tendencies change over time—are not well understood. To help fill this gap in the literature, we used data from two longitudinal studies (Study 1 $N = 107$; Study 2 $N = 1,037$) conducted during the COVID-19 pandemic. We find that greater belief in COVID-19 conspiracy theories at baseline predicts both greater endorsement of a novel real-world conspiracy theory involving voter fraud in the 2020 American Presidential election (Study 1) and increases in generic conspiracist ideation over a period of several months (Studies 1 and 2). Thus, engaging with real-world conspiracy theories appears to act as a gateway, leading to more general increases in conspiracist ideation. Beyond enhancing our knowledge of conspiracist ideation, this work highlights the importance of fighting the spread of conspiracy theories.

## Introduction

In 1998, *abovetopsecret.com* appeared on the internet [1]. Consisting of a few pages devoted to discussing a wide variety of conspiracy theories, this website provided a home for a small number of like-minded people (often colloquially referred to as *conspiracy theorists)* to connect and exchange ideas. More than twenty years later, this site has grown into the largest home for the conspiratorially-minded to discuss topics ranging from plots by the global elite to create a New World Order to alien contact [2, 3]. Although a recent explosion of research on belief in conspiracy theories has greatly advanced our understanding [4], little is known about the variables that can trigger increases in people's general tendency to believe conspiracy theories. What might cause someone to go from having a passing interest in one or two conspiracy theories to more firmly believing a variety of such theories? In the current work, we use longitudinal data to investigate one avenue by which people may become more conspiratorially-minded: through initial "gateway" conspiracy theories that pave the way for greater receptivity to a wide variety of conspiracy theories.

**Data Availability Statement:** Data, syntax, and a full list of measures for Study 1 and syntax specific to the analyses reported in Study 2 are available at

https://osf.io/pb5v9/. To access the publicly-available data and the accompanying materials for Study 2, please visit https://osf.io/v2zur/.

**Funding:** This work was supported by a RAPID grant from the National Science Foundation under Award ID BCS-2031097 (RHF). The funders had no role in study design, data collection and analysis, decision to publish, or preparation of the manuscript.

**Competing interests:** The authors have declared that no competing interests exist.

## A brief review of the literature on conspiracy theory beliefs

Recent events (e.g., January 6[th] Capitol Riot) have brought conspiracy theories and their believers to the forefront of the public's mind [5, 6]. This increased public attention has been mirrored by greater interest from behavioral scientists: the number of publications on conspiracy theory beliefs has increased greatly since the start of the new millennium.

One avenue that researchers have pursued focuses on the extent to which there are indeed stable individual differences in what is referred to as *conspiracist ideation* [7, 8]—the general tendency to believe conspiracy theories. Two broad measurement approaches have been pursued. One presents participants with a number of statements about a variety of popular conspiracy theories involving real-world events (e.g., alternative accounts of the assassination of John F. Kennedy), and asks them to indicate their agreement with each. Studies employing this event-based approach suggest that the constituent items cohere well and conspiracist ideation exists on a continuum ranging from a general tendency to reject conspiracy theories to a general tendency to endorse them [9, 10]. Consistent with this idea, conspiracist ideation—as assessed by these event-based scales—has been shown to correlate *concurrently* with both (1) other real-world conspiracy theories and (2) novel conspiracy theories that have been fabricated by experimenters [11, 12].

A second approach to assessing conspiracist ideation focuses not on specific conspiracy theories involving real-world events, but rather the basic elements that underlie conspiracy theories in general. This approach posits that belief in real-world conspiracy theories originates from a smaller subset of *generic conspiracist beliefs*—decontextualized ideas involving secret plots by powerful groups [7]. For example, conspiracy theories about the assassinations of John F. Kennedy and Princess Diana share the generic conspiracist belief that the government is involved in the murder of prominent citizens. Proponents of measures of generic conspiracist beliefs point to greater consistency in content and broader generality across time and cultures as relative advantages over event-based measures of conspiracist ideation [7]. To create such measures, researchers studied a broad variety of real-world conspiracy theories and distilled the generic conspiracist beliefs that underlie them [13]. Thus, event- and generic conspiracist belief-based measures differ in their level of abstraction.

Importantly, it has been suggested that there are important differences between popular measures of conspiracist ideation [14]. On the one hand, instruments like the Generic Conspiracist Beliefs Scale [7] capture the extent to which people agree with general notions of conspiracy theories (e.g., "Some UFO sightings and rumors are planned or staged in order to distract the public from real alien contact"). On the other hand, instruments like the Conspiracy Mentality Questionnaire [8] tap into the broader ideological mindset of conspiracy mentality by assessing the degree to which people endorse aspects of conspiracist thinking (e.g., "I think that there are secret organizations that greatly influence political decisions"). For simplicity, and given the substantial empirical relation between these two measures [15], we refer to what is measured by each approach as conspiracist ideation—people's general tendency to believe conspiracy theories. Research employing these instruments indicates that conspiracist ideation is highly stable over time, and, most importantly, predicts—again concurrently—belief in conspiracy theories concerning more specific events [see 15 for a review].

Research has also uncovered several predictors of this general tendency to believe conspiracy theories. Prior cross-sectional work has found that greater conspiracist ideation is predicted by higher levels of need for uniqueness [16], higher dispositional levels of anxiety [17], and a more intuitive thinking style [18]—to name a few. Greater narcissism has also been linked to higher levels conspiracist ideation [19]. In one of the relatively few longitudinal

investigations involving conspiracist ideation, researchers found that greater collective narcissism—an exaggerated belief in the superiority of one's in-group—at baseline significantly predicted increases in generic conspiracist beliefs over a four-month period during the 2016 American presidential campaign [20; see 21, 22 for other examples of longitudinal work involving conspiracy theories].

Importantly, however, not everyone who believes a specific conspiracy theory is a conspiracy theorist. Indeed, a majority of people report believing at least one real-world conspiracy theory, yet few people indicate agreement with a majority of conspiracy theories [10]. Thus, belief in a single conspiracy theory is not in and of itself an indication of conspiracist ideation. Consistent with this idea, research suggests that a variety of situational factors can nudge people toward endorsing a given specific conspiracy theory, including a lack of control [23], anxiety [17], and social identity threat [24]. More generally, people seem to be drawn to believing conspiracy theories when they experience threats to existential (e.g., control, security), epistemic (e.g., understanding, accuracy), and social (e.g., self and group image) motives [4, 25].

## The gateway conspiracy hypothesis

Little research has examined the psychological triggers that promote changes in conspiracist ideation. For instance, how might someone who initially has no particularly strong affinity for conspiracy theories begin to move toward becoming a conspiracy theorist? Leveraging insights from the literature on person-environment transactions, the current work seeks to begin to answer this question. Specifically, prior research suggests that even though personality characteristics are, by definition, largely stable over time, they can and do change under certain circumstances [26–28]. Such shifts in personality characteristics are believed to depend on momentary changes in people's thoughts, feelings, and behaviors that are triggered by their perceptions of events. If maintained over a sufficiently long period of time, these momentary changes can crystallize into enduring characteristics [29, 30].

The central thesis of this research is that believing in a specific conspiracy theory can act as a trigger for increases in conspiracist ideation, paving the way for greater receptivity to conspiracy theories in general—an idea we refer to as the *gateway conspiracy hypothesis*. To test this hypothesis, we used data from multiple longitudinal studies that aimed to understand the factors that shaped people's responses to the COVID-19 pandemic [31–33].

## Conspiracy theories and the COVID-19 pandemic

The widespread damage created by the COVID-19 pandemic has heightened people's anxiety, uncertainty, feelings of powerlessness, and has created friction between members of different social groups [34]. As alluded to above, such conditions represent a near perfect recipe for the proliferation of conspiracy theories [4, 25]. Indeed, many conspiracy theories about COVID-19 have emerged [35]. The current work focuses on two common COVID-19 conspiracy theories that have been the subject of prior research—that COVID-19 was intentionally created and released for evil purposes and is being intentionally portrayed as more dangerous than it actually is [36].

A recent study shed light on the predictors of belief in these very same COVID-19 conspiracy theories [37]. Greater belief in the conspiracy theories was independently predicted by higher conspiracist ideation—that is, a greater general tendency to believe conspiracy theories—as well as personally suffering greater negative economic consequences as a result of the pandemic—a situational threat. Moreover, these two predictors interacted such that the relation between the experience of negative economic consequences and belief in COVID-19

conspiracy theories was strongest among those highest in conspiracist ideation, further documenting the significant relevance of generic conspiracy beliefs. Importantly for the present argument, however, the relation between negative economic consequences and conspiracy theories about COVID-19 remained evident even among those relatively low in conspiracist ideation. Indeed, at one *SD* below the mean on the conspiracist ideation scale, the relation was just shy of traditional statistical significance ($p = .06$). In other words, real-world situational pressures can nudge people toward believing specific conspiracy theories, in this case regarding COVID-19—even among those generally less inclined to initially have generic conspiracist beliefs.

## Current research

What might be the downstream consequences of coming to believe these specific conspiracy theories for an individual's general tendency toward conspiracist ideation? One possibility is that people's level of conspiracist ideation is relatively insensitive to environmental pressures, and as a result, coming to believe specific conspiracy theories has no effect on conspiracist ideation over time. Alternatively, it is possible that believing COVID-19 conspiracy theories increases people's belief in conspiracy theories with content congruent to that of pandemic-related conspiracy theories (e.g., the deliberate release of viruses), but not in other kinds of conspiracy theories. After all, the items that comprise the popular measures of conspiracist ideation concern such possibilities as contact with aliens and assassinations of high-profile citizens perpetrated by local governments—none of which show any clear relevance to the circumstances surrounding the pandemic.

On the other hand, it is possible that people's level of conspiracist ideation will change as a function of endorsement of COVID-19 conspiracy theories. Specifically, we propose that once participants come to believe specific conspiracy theories, changes in their thoughts, feelings, and behaviors brought on by endorsement of the conspiracy theories will translate into greater general receptivity to conspiracy theories. To test this idea, we measured baseline belief in COVID-19 conspiracy theories and captured subsequent conspiracist ideation in two ways in Study 1. First, we assessed belief in an unrelated conspiracy theory about fraud in the 2020 American Presidential election, which we administered in a follow-up wave of data collection six months later. This measure allowed us to capture the development of belief in a novel real-world conspiracy theory. We hypothesized that greater belief in COVID-19 conspiracy theories at baseline would be associated with greater belief in the claim of voter fraud in the election at follow-up, even when statistically controlling for potential third variables like baseline conspiracist ideation and political orientation.

Second, we assessed conspiracist ideation at both baseline and at our six-month follow-up using the Generic Conspiracist Beliefs Scale [7]—a popular and well-validated measure of the construct. We hypothesized that greater belief in COVID-19 conspiracy theories at baseline would predict increases in conspiracist ideation over a six-month period.

In Study 2, we sought to provide an additional test of the gateway conspiracy hypothesis by using publicly-available data collected by the COVID-19 Psychological Research Consortium (C19PRC) [33]. The C19PRC conducted a large-scale, multi-wave study focused on understanding reactions to the COVID-19 pandemic among residents of the United Kingdom over a series of months [38]. Included among a variety of other instruments was a measure of belief in COVID-19 conspiracy theories and repeated administrations of a measure of conspiracist ideation—the Conspiracy Mentality Questionnaire [8]—that allowed us to test our hypothesis. Importantly, this replication effort included a larger sample collected in different sociopolitical context using different measures of the key constructs.

## Study 1

### Methods

**Participants.**   All participants provided informed written consent prior to initiating the study. The Institutional Review Board at The Ohio State University approved all study procedures (IRB: 2020B0129). Participants were recruited via the Mechanical Turk (MTurk) platform. Although MTurk samples tend not to be nationally representative, they are considerably more demographically and ideologically diverse than the college student or convenience samples often used in other psychological research [39]. Moreover, MTurk provides geographic diversity within the United States, allowing researchers to reach people in all 50 states. Importantly, prior research suggests that MTurk samples perform similarly to samples drawn from other sources across many tasks and measures [40].

United States-based MTurk workers with 500+ approved HITs and a minimum approval rate of 95% were eligible to participate. Five hundred and one MTurk workers participated in a 10-minute survey in exchange for $1.00 on June 9, 2020. The target sample size for the baseline wave was selected to ensure stable estimates of correlation coefficients [41] and a reasonably large sample for longitudinal analyses.

Four hundred and seventy-five workers who participated in the baseline wave agreed to be contacted for a later wave of data collection. On December 7, 2020, these individuals were invited to participate in the follow-up, and were given two weeks to do so. In total, 107 participants (Gender: 56 female, 51 male; Age [years]: $M = 40.2$, $SD = 14.0$) took part in the six-month follow-up session. As an additional assurance of data quality, an attention/comprehension check was embedded within the study. No participants failed to respond correctly. A sensitivity analysis revealed that we were 80% powered to detect a small effect, $f^2 = 0.076$ [42].

Although our attrition rates were high, they were reasonably close to those observed in other longitudinal MTurk studies [43], and consistent with the rapid turnover of the MTurk worker population [44]. Furthermore, the central variables in this study were not related to attrition, and several steps that we took to examine the impact of attrition [45] provided assurance that its role was minimal (see S1 File).

### Materials

All measures in Study 1 were presented using the Qualtrics survey platform. All relevant measures, manipulations, and exclusions are reported. For data, syntax, and a full list of measures, please visit https://osf.io/pb5v9/?view_only=85cd0287914b498184adaf84840763dc.

**Baseline.**   *Belief in COVID-19 conspiracy theories.* We assessed belief in COVID-19 conspiracy theories by adapting two items generated by Imhoff & Lamberty [36]. Specifically, we asked participants to report the extent to which they believed that (1) the coronavirus was human-made ("COVID-19 was intentionally brought into the world for dark purposes") and that (2) the severity of COVID-19 was exaggerated ("COVID-19 is intentionally presented as dangerous in order to mislead the public") on a response scale with the endpoints *-3. Strongly disagree* and *+3. Strongly agree*. To create a single index of belief in COVID-19 conspiracy theories, we averaged these two items into a single composite score, $r = 0.78$, $p < 0.001$. Importantly, however, our results and conclusions are not substantively altered if the two items are examined separately (see S1 File).

*Political orientation.* Political orientation was measured using one item ("Please select the scale point that best reflects your political orientation") rated on a seven-point response scale ranging from *Extremely liberal* to *Extremely conservative*, where higher values indicate greater conservatism.

*Conspiracist ideation.* Conspiracist ideation at baseline was assessed using a shortened 10-item version of the Generic Conspiracist Beliefs Scale [7]. We elected to employ a trimmed version of the original scale to reduce participant burden. Within each factor of the original scale, the two items with the highest item-total correlations based on earlier studies were selected for inclusion in the study. Each item (e.g., "Some UFO sightings and rumors are planned or staged in order to distract the public from real alien contact"; "New and advanced technology which would harm current industry is being suppressed") was rated on a five-point response scale ranging from *Definitely not true* to *Definitely true*, with higher values indicating greater generic conspiracist beliefs. This trimmed scale maintained very good internal consistency, $\alpha = 0.93$.

**Follow-up.** *Belief in election fraud conspiracy theory.* We first measured conspiracist ideation in the follow-up wave using reports of belief in a novel real-world conspiracy theory unrelated to COVID-19. Specifically, participants reported the extent to which they believed that there was voter fraud in the 2020 American Presidential election ("In your view, to what extent was voter fraud involved in the Presidential election?") on a scale ranging from *0. Not at all* to *5. Very much so.*

*Conspiracist ideation.* We also measured conspiracist ideation at follow-up by once again administering a trimmed version of Generic Conspiracist Beliefs scale [7]. In this wave of data collection, we employed a five-item version of the original scale to further minimize participant burden. Consistent with the approach described above, we created this shortened scale by selecting the item with the highest item-total correlation within each of the five factors. This trimmed scale once again maintained very good internal consistency, $\alpha = 0.89$.

## Results

A correlation matrix of all the key variables in this study is presented in Table 1. The data were analyzed via Ordinary Least Squares regression using the IBM SPSS Statistics 27 software. All variables were standardized prior to being entered into the model.

First, we tested whether belief in COVID-19 conspiracy theories at baseline prospectively predicted the endorsement of a novel, unrelated conspiracy theory. Consistent with our hypothesis, greater belief in COVID-19 conspiracy theories at baseline predicted greater belief in the subsequent conspiracy theory that there was fraud in the 2020 American Presidential election, $\beta = 0.65$ (95% CI: 0.49, 0.80), $t(105) = 8.76$, $p < 0.001$.

However, both initial conspiracist ideation and political orientation serve as plausible alternative explanations for the relation above. That is, belief in COVID-19 conspiracy theories could predict endorsement of the novel voter fraud conspiracy theory either because endorsement of both reflect people's baseline level of conspiracist ideation or because both conspiracy theories under study tended to be more popular among American conservatives. To rule out these alternative explanations, we conducted a hierarchical linear regression in which we predicted belief in voter fraud during the Presidential election at follow-up from baseline conspiracist ideation and political orientation in an initial step, and then entered belief in COVID-19 conspiracy theories as an additional predictor. As expected, greater belief that there was voter fraud in the election was significantly predicted by higher levels of conspiracist ideation, $\beta = 0.19$ (95% CI: 0.13, 0.35), $t(104) = 2.48$, $p = 0.015$, and greater conservatism, $\beta = 0.61$ (95% CI: 0.46, 0.76), $t(104) = 8.21$, $p < 0.001$. Most importantly, even when statistically controlling for these variables, belief in COVID-19 conspiracy theories significantly predicted belief in election voter fraud, $\beta = 0.50$ (95% CI: 0.22, 0.77), $t(104) = 5.29$, $p < 0.001$.

Although these analyses provided initial support for the gateway conspiracy hypothesis, a more stringent test involves predicting change in generic conspiracist ideation from baseline

**Table 1. Correlation matrix of key variables from Study 1.**

| Variable | 1. | 2. | 3. | 4. | 5. |
|---|---|---|---|---|---|
| 1. Belief in COVID-19 conspiracy theories | - | | | | |
| 2. Conspiracist ideation (Baseline) | .68** | - | | | |
| 3. Political orientation (higher numbers = more conservative) | .49** | .30* | - | | |
| 4. Conspiracist ideation (Follow-up) | .78** | .84** | .44** | - | |
| 5. Belief in voter fraud conspiracy theory | .65** | .37** | .67** | .50** | - |

*Note.* N = 107.

* indicates $p = .001$.

** indicates $p < .001$.

endorsement of COVID-19 conspiracy theories. We regressed conspiracist ideation at follow-up first on conspiracist ideation at baseline and then added belief in COVID-19 conspiracy theories at baseline to the model. In the first step, conspiracist ideation assessed at baseline significantly predicted conspiracist ideation six months later, $\beta = 0.84$ (95% CI: 0.75, 0.93), $t(104) = 15.98$, $p < 0.001$—as was expected. The second step revealed that, above and beyond this effect of initial conspiracist ideation, greater belief in COVID-19 conspiracy theories at baseline significantly predicted increases in conspiracist ideation over a six-month period, $\beta = 0.38$ (95% CI: 0.24, 0.51), $t(104) = 6.20$, $p < 0.001$. Importantly, this was no trivial effect: belief in COVID-19 conspiracy theories at baseline accounted for approximately 8% of the variance in conspiracist ideation at follow-up.

We supplemented these results with several follow-up analyses. First, we found that belief in COVID-19 conspiracy theories is also a significant predictor of increases in conspiracist ideation when using only those five items of the Generic Conspiracist Beliefs Scale that were included at both timepoints, $\beta = 0.43$ (95% CI: 0.28, 0.56), $t(104) = 6.81$, $p < 0.001$. Moreover, we complemented the lagged regression approach above with an analysis that employed a conspiracist ideation change score as the outcome variable, i.e., the difference in the five-item scale score at follow-up compared to baseline. Consistent with the results reported above, greater belief in COVID-19 conspiracy theories at baseline predicted increases in conspiracist ideation over time, $\beta = 0.28$ (95% CI: 0.097, 0.44), $t(104) = 2.96$, $p = 0.004$. Lastly, just as is to be expected on the basis of the very strong internal consistency we observed among the five Generic Conspiracist Belief items that we employed at follow-up ($\alpha = 0.89$), these lagged-regression effects were evident to at least some degree on each and every item when tested individually ($p$'s ranging from $< 0.001$ to 0.15; see S1 File).

## Discussion

The results of this study provide converging lines of evidence in support of the gateway conspiracy hypothesis. Believing COVID-19 conspiracy theories predicted belief in the novel conspiracy of voter fraud in the 2020 American Presidential election six months later, even when statistically controlling for potential confounds. Moreover, greater belief in COVID-19 conspiracy theories at baseline predicted *increases* in conspiracist ideation over a six-month period. Together, these findings suggest that engaging with real-world conspiracy theories can lead to increases in the general tendency to believe conspiracy theories over time.

In Study 2, we used a publicly-available dataset collected in the United Kingdom (UK) to conceptually replicate the relation between belief in COVID-19 conspiracy theories and increases in conspiracist ideation reported in Study 1. Given that Study 2 uses a larger dataset collected in a different sociopolitical context and employs alternative measures of the

constructs of interest, results consistent with those of Study 1 would offer strong converging evidence and, hence, increase confidence in the effects observed in Study 1.

## Study 2

### Methods

**Participants.**   English-speaking adults residing in the UK were recruited for participation via Qualtrics [38]. The current analysis focused on the measurement occasions that included measures of our constructs of interest, which were the first (conducted between March 23–28, 2020), second (April 22-May 1, 2020), and fourth waves (November 25-December 22, 2020) of data collection from the larger C19PRC study.

The initial wave of data collection consisted of 2,025 participants. The available sample for our analyses involved 1,037 participants (Gender: 495 female, 541 male, 1 preferred not to say; Age [years]: $M$ = 51.0, $SD$ = 14.5). A sensitivity analysis revealed that we were 80% powered to detect a small effect, $f^2$ = 0.006 [42]. Importantly, attrition in this study was predominantly predicted by demographic rather than psychological variables [46]. More to the point, neither conspiracist ideation nor belief in COVID-19 conspiracy theories significantly related to attrition, despite the statistical power provided by the large sample size, and additional analyses revealed that it played no significant role in shaping the results (see S1 File).

### Materials

To access the publicly-available data and the accompanying study materials for the C19PRC study, please visit https://osf.io/v2zur/. For syntax specific to the analyses reported here, see https://osf.io/pb5v9/?view_only=85cd0287914b498184adaf84840763dc.

**Wave 1.**   *Conspiracist ideation.* Conspiracist ideation assessed via the Conspiracy Mentality Questionnaire [8]. This instrument consists of five generic statements that capture a conspiratorial view of the world (e.g., "I think that many very important things happen in the world, which the public is never informed about"), α = 0.86. Participants report the extent to which they agree with each item using an 11-point scale anchored by the endpoints *Certainly not– 0%* and *Certainly– 100%*.

**Wave 2.**   *Belief in COVID-19 conspiracy theories.* Mirroring Study 1, participants reported the degree to which they believed that COVID-19 was human-made ("Covid-19 was developed in a lab in Wuhan, China") and that the severity of the virus has been misrepresented ("Coronavirus is actually no more dangerous than the common flu"). Each item was rated using a slider scale that ranged from *Do not believe at all 0%* to *Completely believe 100%*. Consistent with Study 1, we combined these two items into a single index representing people's belief in COVID-19 conspiracy theories, $r$ = 0.25, $p < 0.001$. As before, our results and conclusions are not substantively altered if the two items are examined separately (see S1 File).

**Wave 4.**   *Conspiracist ideation.* Conspiracist ideation was again measured using the five-item Conspiracy Mentality Questionnaire, α = 0.89 [8].

### Results

Table 2 presents a correlation matrix of the key variables in this study. The data were analyzed via Ordinary Least Squares regression using the IBM SPSS Statistics 27 software. All variables were standardized prior to being entered into the model.

We conducted a hierarchical linear regression in which we predicted conspiracist ideation at Wave 4 from a person's score on this same variable at Wave 1, and then added belief in COVID-19 conspiracy theories as a predictor in a subsequent step. The results revealed that

**Table 2. Correlation matrix of key variables from Study 2.**

| Variable | 1. | 2. | 3. |
|---|---|---|---|
| 1. Conspiracist ideation (Wave 1) | - | | |
| 2. Belief in COVID-19 conspiracy theories (Wave 2) | .22* | - | |
| 3. Conspiracist ideation (Wave 4) | .60* | .29* | - |

*Note.* $N = 1,037$.

* indicates $p < .001$.

conspiracist ideation at Wave 1 significantly predicted conspiracist ideation at Wave 4, $\beta = 0.60$ (95% CI: 0.55, 0.65), $t(1035) = 24.26$, $p < 0.001$. Most importantly, our central prediction was supported: greater belief in COVID-19 conspiracy theories predicted an increase in conspiracist ideation over time, $\beta = 0.16$ (95% CI: 0.11, 0.21), $t(1034) = 6.40$, $p < 0.001$.

Once again, we supplemented this lagged-regression approach with one that employed a change score (Wave 4 conspiracist ideation–Wave 1 conspiracist ideation) as the outcome variable, and found that greater belief in COVID-19 conspiracy theories similarly predicted increases in conspiracist ideation, $\beta = 0.12$ (95% CI: 0.05, 0.18), $t(1035) = 3.79$, $p < 0.001$. Moreover, this lagged-regression effect was detected to some degree on all items when tested singly ($p$'s ranging from $< 0.001$ to $0.16$; see S1 File)—replicating the results in Study 1.

## Discussion

In Study 2, we leveraged a large publicly-available dataset to provide an additional test of the gateway conspiracy hypothesis. Mirroring the findings reported previously, greater belief in COVID-19 conspiracy theories predicted increases in conspiracist ideation over a period of several months. These results represent an independent replication of the gateway conspiracy hypothesis using a larger sample collected in a different sociopolitical context and with different operationalizations of the conceptual variables of interest.

## General discussion

Across two studies, we find consistent support for the gateway conspiracy hypothesis. In Study 1, belief in COVID-19 conspiracy theories prospectively predicted endorsement of a novel conspiracy theory involving fraud in the 2020 American Presidential election, even when controlling for baseline conspiracist ideation and political orientation. Moreover, greater belief in COVID-19 conspiracy theories predicted increases in conspiracist ideation over a period of several months—a finding replicated in Study 2 using a large UK sample. Thus, the results suggest that engaging with real-world conspiracy theories can pave the way for subsequent increases in people's general tendency to believe conspiracy theories. Numerous prior cross-sectional studies have shed light on the suite of variables that relate to conspiracist ideation at a given point in time, yet attention has rarely been devoted to studying changes in conspiracist ideation over time [20]. By documenting the gateway conspiracy effect in a prospective manner, the current research begins to shed light on this neglected question of what factors can trigger changes in conspiracist ideation occur, and points to engagement with specific conspiracy theories as an avenue for increase in this regard.

One obvious limitation to the present research concerns the attrition that was evident in the two longitudinal studies. Although attrition across the six-month period over which Study 1 was conducted was admittedly high, the retention rate of 23% was comparable to other longitudinal studies conducted before and during the pandemic [43, 47]. Despite the relatively

small size of the resulting sample, the longitudinal effect that we observed regarding the relation between COVID-19 conspiracy beliefs and subsequent conspiracist ideation was very evident statistically, accounting for an additional 8% of the variance beyond that accounted for by initial conspiracist ideation. The retention rate in Study 2 was substantially higher (51%) and, most importantly, the study involved a far larger sample for our prospective data analyses, the results of which replicated the findings of Study 1. Notably, the strength of the statistical evidence in support of the gateway conspiracy hypothesis in both studies was compelling. The *p*-values associated with each key test of the hypothesis were well below what even the most cautious social scientists regard as evidence for a significant new effect [48].

Attrition does pose an issue with respect to the extent to which it is appropriate to generalize from the present findings. Technically speaking, the findings should not be generalized beyond individuals who are willing to maintain participation in multiple surveys across a 6–9-month time period. However, this limitation does not pose a threat to the interval validity of the longitudinal results. Importantly, it is not at all clear how attrition can offer a basis for an alternative explanation for the findings. This is especially so in light of the analyses detailed in S1 File. Attrition was not related to conspiracist ideation at either time point, nor to belief in COVID-19 conspiracy theories, in either study. In addition, those analyses highlight the comparability of the subsample who participated in the follow-up session to the full sample from the initial survey. To sum up, then, although the rate of attrition present in these studies is not ideal, there is no indication that attrition threatens the validity of our findings.

Another limitation of the current work concerns the manner in which our measures of interest were distributed through the various waves of data collection. Having measures of belief in COVID-19 conspiracy theories at the earlier timepoints and conspiracist ideation at both the earlier and later timepoints allowed us to predict changes in conspiracist ideation from belief in COVID-19 conspiracy theories. However, the lack of a measure of belief in COVID-19 at the later timepoints prevents us from using cross-lagged panel models to also estimate the effect of conspiracist ideation on change in belief in COVID-19 conspiracy theories. Future research should seek to address this limitation. Such work may reveal that while belief in specific conspiracy theories (analogous to those studied here involving COVID-19) predicts future conspiracist ideation, conspiracist ideation is more weakly related to future belief in specific conspiracy theories. However, it also seems possible that while initial belief in specific conspiracy theories predicts changes conspiracist ideation, initial conspiracist ideation also predict increases in belief in specific conspiracy theories. That is, there may be a bidirectional relation between belief in specific conspiracy theories and conspiracist ideation. Indeed, previous research suggests that, by its nature, conspiracist ideation is robustly associated with endorsement of specific event-related conspiracy theories [15, 37]. Such bidirectional causality would point to a reinforcing cycle in which belief in specific conspiracy theories and conspiracist ideation amplify each other.

One interesting issue to ponder is the degree to which the findings reported here will readily generalize to contexts that differ from that of the current study. There is anecdotal evidence that previous large-scale tragedies like the 9/11 terrorist attacks led to the development of conspiracy theories that acted as gateways for some members of the population. For instance, when a post on the aforementioned conspiracy theory website *abovetopsecret.com* asked members to share how they became "conspiracy nuts," some users reported being lured down the rabbit hole after engaging with conspiracy theories related to the 9/11 attacks [49]. Moreover, a former conspiracy theorist recounted his descent into the world of conspiracy theories in the following way: "I came across [the 9/11 conspiracy theory documentary] *Loose Change* and Alex Jones circa 2006 and it sort of just spiraled from there" [50]. We contend that while socially-significant events like the 9/11 terror attacks and the COVID-19 pandemic are

particularly likely to give rise to gateway conspiracy theories, all conspiracy theories can serve this role. From our perspective, when specific conspiracy theories lure people in, they set off a sequence of psychological processes (e.g., reduced trust in institutions) that guide people in the direction of being receptive toward a wider variety of conspiracy theories, and this pattern should not substantively differ according to the specific content of the conspiracy theory.

The evidence provided here in support of the gateway conspiracy hypothesis also has important practical implications. Specifically, it suggests that situational forces that nudge people toward endorsing specific conspiracy theories in the present may have serious downstream effects. A large body of work suggests that situational threats can push people toward believing specific conspiracy theories [4, 23, 25]. In fact, our own prior work found that experiencing negative economic consequences due to the pandemic was associated with greater belief in COVID-19 conspiracy theories—even among those reporting low levels of generic conspiracist beliefs [37]. Such findings suggest that situational forces can be sufficiently strong so as to push people toward endorsing specific conspiracy theories, even people who tend not to be very conspiratorially-minded initially. These very nudges can then lead to subsequent increases in more generic conspiracist ideation.

Moreover, research suggests that conspiracist ideation is associated with a variety of negative outcomes, for both the believers and those that they encounter, including prejudice, dangerous health behaviors, political disengagement, and radicalization [4]. From a practical perspective, then, the present findings bring a new sense of importance to stopping the spread of COVID-19 conspiracy theories: Not only do COVID-19 conspiracy theories threaten lives and economies in the present, they may also create problems down the road by leading to heightened conspiracist ideation. Policymakers would be wise to consult the research that has tested strategies by which belief in conspiracy theories can be blunted [51].

Future research should address the mechanisms through which belief in specific conspiracy theories triggers increases in conspiracist ideation. A loss of trust in authorities is one possible mechanism [52]. Specifically, once people believe a specific conspiracy theory that implicates scientists or government officials in wrongdoing, their trust in those authorities will be violated, making it more likely that they believe other novel conspiracy theories in which those individuals are said to be involved, even if those novel conspiracy theories are unrelated to the original. Moreover, a social component may play a role in driving greater receptivity to conspiracy theories. For example, someone who comes to believe that COVID-19 is a hoax may be drawn to a conspiracy theory website like that mentioned in the introduction. Once in that environment, they would be exposed to a wide variety of conspiracy theories and encouraged to discuss them with individuals who are inclined to believe conspiracy theories.

Beyond investigating potential mechanisms for the increases in conspiracist ideation documented here, future research should consider the forces associated with decreases in conspiracist ideation. Stated differently, while the current research sheds light on a psychological factor that pushes people further down the conspiracy theory rabbit hole, what might pull people out of the rabbit hole? By tackling such questions, ones involving the factors associated with both growth and decay, we can gain a fuller understanding of the dynamics of conspiracist ideation.

## Supporting information

**S1 File. A collection of supplemental analyses.**
(DOCX)

## Acknowledgments

We would like to thank Jonathan L. Stahl for providing helpful feedback on an earlier version of this manuscript.

## Author Contributions

**Conceptualization:** Javier A. Granados Samayoa.

**Data curation:** Javier A. Granados Samayoa.

**Formal analysis:** Javier A. Granados Samayoa.

**Funding acquisition:** Russell H. Fazio.

**Investigation:** Javier A. Granados Samayoa, Courtney A. Moore, Benjamin C. Ruisch, Shelby T. Boggs, Jesse T. Ladanyi, Russell H. Fazio.

**Methodology:** Javier A. Granados Samayoa, Courtney A. Moore, Benjamin C. Ruisch, Shelby T. Boggs, Jesse T. Ladanyi, Russell H. Fazio.

**Project administration:** Javier A. Granados Samayoa, Courtney A. Moore, Benjamin C. Ruisch, Shelby T. Boggs, Russell H. Fazio.

**Resources:** Russell H. Fazio.

**Software:** Courtney A. Moore, Benjamin C. Ruisch.

**Supervision:** Russell H. Fazio.

**Validation:** Russell H. Fazio.

**Writing – original draft:** Javier A. Granados Samayoa.

**Writing – review & editing:** Javier A. Granados Samayoa, Courtney A. Moore, Benjamin C. Ruisch, Shelby T. Boggs, Jesse T. Ladanyi, Russell H. Fazio.

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
