## [Decision Letter · Decision Letter 0]

8 Sep 2022

PONE-D-22-16710A gateway conspiracy? Belief in COVID-19 conspiracy theories prospectively predicts greater conspiracist ideationPLOS ONE

Dear Dr. Fazio,

Thank you for submitting your manuscript to PLOS ONE. After careful consideration, we feel that it has merit but does not fully meet PLOS ONE’s publication criteria as it currently stands. Therefore, we invite you to submit a revised version of the manuscript that addresses the points raised during the review process. I would like to take this opportunity to thank the reviewers for their careful reading of the manuscript. I would be grateful if you could consider all the points they raise in a revision of the manuscript.

Reviewer 1 raises an interesting point about cross-lagged longitudinal analyses. From my reading of the manuscript I thought there was strong control of baseline measures of general conspiracy beliefs in the analyses predicting general conspiracy outcomes. However, I agree with the reviewer that cross-lagged paths might also be interesting. For example, finding that generic conspiracy beliefs are not such strong predictors of change in COVID conspiracy beliefs over time might strengthen your interpretation that COVID conspiracy is a gateway. Therefore, if you have outcome measures of COVID conspiracy beliefs then I would be grateful if you could consider including analyses of these outcomes in the manuscript. If you do not have measures of COVID conspiracy beliefs at outcome then please consider discussing this as a limitation of the study.

Following my reading I would also be grateful if you could add citations to Introduction paragraph 1 support the points made.

We look forward to receiving your revised manuscript.

Kind regards,

Richard Rowe

Academic Editor

PLOS ONE

Journal Requirements:

Reviewers' comments:

Reviewer's Responses to Questions

**Comments to the Author**

1. Is the manuscript technically sound, and do the data support the conclusions?

Reviewer #1: Yes

Reviewer #2: Yes

2. Has the statistical analysis been performed appropriately and rigorously? 

Reviewer #1: I Don't Know

Reviewer #2: Yes

3. Have the authors made all data underlying the findings in their manuscript fully available?

Reviewer #1: Yes

Reviewer #2: Yes

4. Is the manuscript presented in an intelligible fashion and written in standard English?

Reviewer #1: Yes

Reviewer #2: Yes

5. Review Comments to the Author

Reviewer #1: This manuscript comprises of two longitudinal studies where the author (s) have investigated how COVID-19 conspiracy theories can predict other conspiracy beliefs several months later. In two studies, the author (s) find support for their predictions. The paper is written-well and the findings are timely. The literature has a dearth of longitudinal designs, so the paper will be welcomed by the scholarly community. I do have some comments that the author (s) may wish to consider:

-It’ll be good to include the sample sizes of the studies within the abstract (e.g., “we used data from longitudinal studies (Study 1 N = xxx, Study 2 N = xxxx) conducted during the COVID-19 pandemic”).

-I am not an expert in analysing longitudinal designs. However, I did wonder why cross-path analyses were not run for the data – that is, controlling for Time 1 and Time 2 measures. I trust another reviewer and/or editor will be able to provide a critical eye on these analyses. Is it also worth including correlational tables in the main body?

-Is it possible to test the alternative predictions – that is, generic conspiracy beliefs predict specific COVID-19 conspiracy beliefs? Showing that this alternative is not plausible (or as strong) feel important.

-Would these findings extend to other conspiracy beliefs, or are they specific to these measured events? I can understand why the election conspiracy theory is novel. However, this conspiracy theory has existed for quite some time, but the difference is the political party that loses argues that the election is a fraud (see some of the work by Joe Uscinski on political losers). Thus, is this theory truly novel?

Reviewer #2: Thank you for the opportunity to review the manuscript "A gateway conspiracy? Belief in COVID-19 conspiracy theories prospectively predicts greater conspiracist ideation". This work was analysed appropriately and the findings were interpreted fairly. I would like to recommend some minor points before this can be accepted for publication, which I have listed below:

1) Your definitions of types of conspiracy beliefs

There seems to me to be some conflation between different conspiracy beliefs measures. Specifically, I believe you seem to conflate belief in general notions of conspiracies (GCB), conspiracy mentality, and conspiracist ideation. While there does appear to be overlap between these measures, I would recommend some changes in how you describe the measures used. While it does not change interpretation of your hypotheses, the GCB certainly measures belief in general notions of conspiracies, whereas conspiracy mentality is a distinct political attitude used as a proxy for the tendency to be susceptible to many conspiracy narratives, and conspiracist ideation is usually conceptualised as an overlap between the two. You may disagree with some of this, but ultimately I would prefer to see more nuanced consideration of the definitions used to describe these different constructs.

2) Failure to mention your extension of the monological belief system

Much recent work has called the specific tenets of the monological account into question. Your model extends the monological belief model and makes more specific predictions about the self-reinforcing nature of conspiracy beliefs. I would recommend mentioning how your hypothesis extends this model with consideration of how it takes more recent findings into account.

6. PLOS authors have the option to publish the peer review history of their article (what does this mean?). If published, this will include your full peer review and any attached files.

Reviewer #1: No

Reviewer #2: No

---

## [Author Response · Author response to Decision Letter 0]

13 Sep 2022

Please note: A more readable copy of this "Response to Reviewers" has been uploaded.

Response to reviewers

We would like to thank the editor and reviewers for their favorable reactions to our manuscript and for taking the time to provide helpful comments. We have carefully considered each comment, and have made multiple revisions to the manuscript. Our manuscript has greatly benefited from your feedback. We reproduce each comment offered in the action letter below in blue font, and provide responses in this black font.

Editor:

1. Reviewer 1 raises an interesting point about cross-lagged longitudinal analyses. From my reading of the manuscript I thought there was strong control of baseline measures of general conspiracy beliefs in the analyses predicting general conspiracy outcomes. However, I agree with the reviewer that cross-lagged paths might also be interesting. For example, finding that generic conspiracy beliefs are not such strong predictors of change in COVID conspiracy beliefs over time might strengthen your interpretation that COVID conspiracy is a gateway. Therefore, if you have outcome measures of COVID conspiracy beliefs then I would be grateful if you could consider including analyses of these outcomes in the manuscript. If you do not have measures of COVID conspiracy beliefs at outcome then please consider discussing this as a limitation of the study.

Thank you for bringing this important point to our attention. We are also in agreement with Reviewer 1 regarding the value of cross-lagged panel models in clarifying the nature of the relation between belief in COVID-19 conspiracy theories and conspiracist ideation. Unfortunately, however, neither Study 1 nor Study 2 include measures of belief in COVID-19 conspiracy theories at the later timepoint. Thus, we are unable to apply a cross-lagged panel model to the data. However, we also are in agreement that this issue merits discussion in the manuscript. We have added text in the general discussion section (p. 22) that points to this issue as a limitation of the current research, and explores different possible outcomes of such cross-lagged path models.

2. Following my reading I would also be grateful if you could add citations to Introduction paragraph 1 support the points made.

We appreciate the reminder to support the assertions made in the first paragraph of the Introduction. We have now added multiple citations to support our assertions regarding the website abovetopsecret.com, and we direct readers to a thorough review of the literature on belief in conspiracy theories before pointing to the gap in that literature that the current research seeks to fill.

We would also like to note that we believe we have now complied with all formatting requirements for the journal.

Reviewer #1:

It’ll be good to include the sample sizes of the studies within the abstract (e.g., “we used data from longitudinal studies (Study 1 N = xxx, Study 2 N = xxxx) conducted during the COVID-19 pandemic”).

We appreciate the reviewer’s suggestion, and see value in specifying the sample size of each study in the abstract. We have now inserted text on p. 2 that reports the sample size for each study.

I am not an expert in analysing longitudinal designs. However, I did wonder why cross-path analyses were not run for the data – that is, controlling for Time 1 and Time 2 measures. I trust another reviewer and/or editor will be able to provide a critical eye on these analyses. Is it also worth including correlational tables in the main body?

Is it possible to test the alternative predictions – that is, generic conspiracy beliefs predict specific COVID-19 conspiracy beliefs? Showing that this alternative is not plausible (or as strong) feel important.

We thank the reviewer for raising this important issue. Given that neither of our studies includes a measure of belief in COVID-19 conspiracy theories at the final timepoint, we are unable to use cross-lagged panel models to model the influence of conspiracist ideation on change in belief in COVID-19 conspiracy theories over time. We have added text on p. 22 pointing to this as a limitation of the study, and discuss different possible outcomes of such analyses.

In addition, we agree with the suggestion to add correlation tables in the main text. These tables have been added on pgs. 13 and 19.

Would these findings extend to other conspiracy beliefs, or are they specific to these measured events? I can understand why the election conspiracy theory is novel. However, this conspiracy theory has existed for quite some time, but the difference is the political party that loses argues that the election is a fraud (see some of the work by Joe Uscinski on political losers). Thus, is this theory truly novel?

Thank you for raising this interesting point regarding the generalizability of the effect we report. We believe that our findings extend beyond the specific context in which this research took place, and have added a paragraph on p. 23 to discuss why we believe this to be a general effect.

Regarding the question of whether the election fraud conspiracy theory is truly novel, we agree that conspiracy theories about stolen elections are nothing new in the domain of politics in a general sense. That is, the idea that an opposing political party has stolen an election is not novel in international politics. However, in the American context in which Study 1 took place, Donald Trump’s claims of voter fraud were very much novel and exploded in popularity shortly after the 2020 American presidential elections, as reflected in Google searches for the term “election fraud” in the United States (see Figure 1).

Figure 1. Google searches for the term “election fraud” in the United States of America over the last five years. Retrieved from https://trends.google.com/trends/explore?date=today%205-y&geo=US&q=%22election%20fraud%22.

Reviewer #2:

1) Your definitions of types of conspiracy beliefs

There seems to me to be some conflation between different conspiracy beliefs measures. Specifically, I believe you seem to conflate belief in general notions of conspiracies (GCB), conspiracy mentality, and conspiracist ideation. While there does appear to be overlap between these measures, I would recommend some changes in how you describe the measures used. While it does not change interpretation of your hypotheses, the GCB certainly measures belief in general notions of conspiracies, whereas conspiracy mentality is a distinct political attitude used as a proxy for the tendency to be susceptible to many conspiracy narratives, and conspiracist ideation is usually conceptualised as an overlap between the two. You may disagree with some of this, but ultimately I would prefer to see more nuanced consideration of the definitions used to describe these different constructs.

We thank the reviewer for encouraging us to explore the distinction between the two measures of conspiracist ideation we employ in our research. We have included text on pgs. 4 and 5 that addresses one way to conceptualize this distinction. As suggested, we distinguish between measures of general notions of conspiracies and conspiracy mentality, yet note that we use the term conspiracist ideation to refer to what is captured by both measures given their strong correlations and our broad interest in measuring the general tendency to believe conspiracy theories.

2) Failure to mention your extension of the monological belief system

Much recent work has called the specific tenets of the monological account into question. Your model extends the monological belief model and makes more specific predictions about the self-reinforcing nature of conspiracy beliefs. I would recommend mentioning how your hypothesis extends this model with consideration of how it takes more recent findings into account.

Thank you for bringing this important point to our attention. We are aware of the debate surrounding the monological belief system view in the literature. As we understand it, there remains active discussion about whether the clustering of conspiracy theory beliefs is best accounted for by the idea that conspiracy theory beliefs mutually reinforce each other in an expanding and closed-off network or an underlying general tendency to believe conspiracy theories. Recent work has tried to tease these seemingly-competing explanations apart, and has taken a position that favors the monological view [1]. However, our view is that these explanations are not in competition, but rather complementary and both play a part in giving rise to the clustering of belief in conspiracy theories. Because we do not believe that the current work speaks to the ongoing debate, we prefer to not raise this issue in the manuscript.

 

References

[1] Williams MN, Marques MD, Hill SR, Kerr JR, Ling M. Why are beliefs in different conspiracy theories positively correlated across individuals? Testing monological network versus unidimensional factor model explanations. British Journal of Social Psychology. 2022 Jan 27.

---

## [Editor Report · Decision Letter 1]

19 Sep 2022

A gateway conspiracy? Belief in COVID-19 conspiracy theories prospectively predicts greater conspiracist ideation

PONE-D-22-16710R1

Dear Dr. Fazio,

Many thanks for addressing the reviews of your original submission so carefully. We’re pleased to inform you that your manuscript has been judged scientifically suitable for publication and will be formally accepted for publication once it meets all outstanding technical requirements.

Kind regards,

Richard Rowe

Academic Editor

PLOS ONE
---

## [Editor Report · Acceptance letter]

3 Oct 2022

PONE-D-22-16710R1 

A gateway conspiracy? Belief in COVID-19 conspiracy
theories prospectively predicts greater conspiracist ideation 

Dear Dr. Fazio:

I'm pleased to inform you that your manuscript has been deemed suitable for publication in PLOS ONE. Congratulations! Your manuscript is now with our production department. 

Kind regards, 

on behalf of

Professor Richard Rowe 

Academic Editor

PLOS ONE